# Surveillance of Adverse Events Following Immunization (AEFI) after Third Dose Booster Vaccination with mRNA-Based Vaccine in Universitas Indonesia Hospital Health Personnel

**DOI:** 10.3390/vaccines10060877

**Published:** 2022-05-30

**Authors:** Rakhmad Hidayat, Alyssa Putri Mustika, Fhathia Avisha, Zlatikha Djuliannisaa, Dinisa Diah Winari, Ria Amiliah Putri, Heydi Marizky Lisman, Vandra Davin, Alvina Widhani, Muhammad Hafiz Aini, Meilisa Rahmadani, Novita Dwi Istanti, Astuti Giantini

**Affiliations:** 1Faculty of Medicine, Universitas Indonesia, Depok 16424, Indonesia; apmalyssa@gmail.com (A.P.M.); d.zlatikha@gmail.com (Z.D.); alvina.widhani@gmail.com (A.W.); astutideges@yahoo.com (A.G.); 2Universitas Indonesia Hospital, Depok 16424, Indonesia; fhathiaa@gmail.com (F.A.); dinisawinari@gmail.com (D.D.W.); riaamiliaputri@gmail.com (R.A.P.); heydi.lisman@gmail.com (H.M.L.); vandravd@gmail.com (V.D.); hafizkudo@gmail.com (M.H.A.); meilisa.rahmadani@rs.ui.ac.id (M.R.); novita.dwi71@ui.ac.id (N.D.I.)

**Keywords:** AEFI, booster vaccine, health workers, mRNA-based vaccine

## Abstract

Facing the rising cases of with higher fatalities COVID-19, some countries decided to give the third dose of vaccine as a booster. As of 9 January 2022, 90.31% of health workers in Indonesia have received the third dose vaccine. This study aims to provide an evaluation of adverse events following immunization (AEFI) in a single center in Indonesia to form a basis for ensuring safety for booster administration nationally. A retrospective, cross-sectional study was conducted using an online survey. Demographic data, AEFI complaints, and factors influencing AEFIs were evaluated. In this study, there were a total of 311 subjects were gathered. The most common AEFI symptoms found at onset <24 h to 28 days were pain at the injection site, fever, shoulder pain, and headache. Most of the AEFI severity of <24 h to 28 days post-vaccination was grade 1 (reduced or uninterrupted daily activities). There was a significant correlation between AEFI and several factors, such as the history of drug allergy, exercise after vaccination, age, BMI < 25, history of symptoms after the first and second vaccinations, and history of COVID-19. There was no anaphylactic reaction in this study. Several AEFI should be considered for the third dose of COVID-19 vaccine administration.

## 1. Introduction

Indonesia has the highest incidence of infection and mortality because of COVID-19 in Southeast Asia. The COVID-19 vaccines have been proven to not only reduce mortality, but also prevent symptomatic COVID-19 infection and can reduce the long-term effects of COVID-19 (long COVID) that can affect the quality of life [1]. Following the guideline of the WHO Strategic Advisory Group of Experts on Immunization (SAGE) Roadmap, health workers are included in the priority group to receive the vaccine. From the beginning of the pandemic until 13 January 2022, the number of Indonesian health workers who died from COVID-19 was 2066. Data from the Indonesia Population and Civil Registration Agency on 31 December 2021 shows that there are 567,910 health workers. Facing a wave of rising recent cases with higher fatalities, some countries have given the third dose or booster vaccine. As of 9 January 2022, 90.31% of health workers in Indonesia have received the third dose of the COVID-19 vaccine [1,2].

Beside the expected benefits of the booster vaccine, there is a potential for improved immune response, thus, the adverse events following immunization (AEFI) of this vaccine must be considered [3]. The occurrence of AEFI after COVID-19 vaccination must be monitored. According to the Association of Indonesian Internal Medicine Specialists (PAPDI), there are at least four reasons why AEFI should be monitored. First, no vaccine is a one hundred percent safe and without risk. Second, it is important to know the risks and how to handle such events when they occur. Furthermore, it is crucial to inform people about AEFIs properly to maintain public confidence in the vaccination program. Finally, AEFI monitoring also helps improve service quality as well [4]. The purpose of this study is to provide an evaluation of adverse events following immunization (AEFI) after administering booster vaccine immunization at a single center in Indonesia. Therefore, the data can be used as a basis for ensuring the safety of booster administration nationally.

## 2. Materials and Methods

### 2.1. Study Design and Population

The study design was cross-sectional and conducted from September to October 2021. The researcher conducted the study at one of the COVID-19 center hospitals in Indonesia (Universitas Indonesia Hospital). The target population was the health workers who have received complete COVID-19 vaccinations (most of them got the Sinovac vaccine for the first and second dose) and subsequently received the Moderna booster vaccine in August and September 2021. The inclusion criteria in this study were subjects who had been screened before the administration, including normal vital signs, history of COVID-19 > 1 month, and had received the third vaccination. The researcher made a notification for participation in this study was made via Google Form, with a link distributed through the WhatsApp application. A questionnaire to assess AEFI was prepared in Bahasa Indonesia and was estimated to be completed within 5 min. The questionnaire included an evaluation of AEFI and had to be filled once after 28 days had passed since the booster vaccination.

### 2.2. Data Collection and Analysis

Respondents filled out a questionnaire that consists of personal data and complaints related to AEFI. The questionnaire data were filled out at a minimum of 28 days after the booster dose administration. The personal data in the questionnaire included gender, weight, height, comorbidities/diseases, history of AEFI, history of COVID-19 infection, history of drug allergy, history of food allergy, and history of other allergies. Through a combination of closed and open questions, the researcher asked the complaints about AEFI. The analytical study was conducted on 311 persons. The questionnaire data were entered into a Microsoft Excel sheet. The data were analyzed the data statistically using Microsoft Excel 2019 and SPSS 24. The Kolmogorov–Smirnov test was used to assess the normality of the data. The characteristics of gender, age, comorbidities, BMI, history of allergies, exercise, the history of COVID-19 infection, history of post-COVID-19 vaccination complaints, and the incidence of AEFIs were compared using the chi-square test. Statistical comparisons were performed using the significance level (*p* < 0.05).

### 2.3. Ethical Clearance

This research was approved by The Ethics Committee of Universitas Indonesia Hospital, approval number S-010/KETLIT/RSUI/II/2022 with protocol number 2021-09-099. This research also followed the Declaration of Helsinki guidelines.

## 3. Results

In this study, we classified the degree of AEFI into three groups, which were normal, grade I, and grade II (Table 1). Grade I are the respondents who felt their activities were reduced or not disturbed. Meanwhile, grade II are the respondents who are could not perform daily activities following the vaccine. Most of AEFI effects start in the first 24 h, and increase during 24–48 h. After 48 h, the symptoms decreased.

Table 2 showed that in the first 24 h after the booster vaccination, there were three main complaints from the respondents. Two hundred and sixty-five subjects (85.2%) experienced pain at the injection site, followed by fever 118 subjects (38%), and shoulder pain 112 subjects (36.1%). At 24–48 h after vaccination, there was a decrease in pain at the injection site, from 265 to 240 subjects (77.2%). Meanwhile, there was an increase in the number of respondents who experienced shoulder pain, from 112 to 160 subjects (51.4%). There was also an increase in the number of subjects who experienced headache, from 13 to 104 (33.3%).

Table 3 shows that for <24 h after booster vaccination, AEFI was influenced by a history of drug allergy, no history of exercise after vaccination, age, and BMI, with a *p*-value of <0.05. Meanwhile, 24–48 h after booster vaccination, the presence of AEFI is influenced significantly by a history of complaints after the first and second vaccinations, and no history of exercise after booster vaccination. For 48 h–7 days after booster vaccination, the presence of AEFI was influenced by a history of complaints after the first and second vaccinations and a history of previous COVID-19, with a *p*-value of <0.05. For 7–28 days after booster vaccination, the presence of AEFI was influenced by a history of complaints after the first and second vaccinations, with *p*-value < 0.05. Respondents who had a history of complaints after the first and second vaccination showed an increased risk of experiencing AEFI symptoms for a duration of 7–28 days after booster vaccination (*p* = 0.03).

## 4. Discussion

In this study, the highest prevalence of symptoms < 24 h after booster vaccination was pain at the injection site (85.2%). Then it followed by fever (38%), shoulder pain (36.1%), and headache (4.2%). Kadali et al. stated the common complaints after administration of the mRNA vaccine to health workers were shoulder pain (94.21%), fatigue/weakness (65.74%), headache (59.26%), muscle pain (54.17%), shivering (52.78%), and fever (35.65%) [5]. The study by Hause et al. also showed that the most common local reaction was pain at the injection site (75.9%) [6]. A study conducted by Bardales et al. found that fever was found to be more frequent after the second dose vaccination (29.5% subjects), compared to the first dose [7]. Jackson et al. showed that almost all subjects after the mRNA vaccine experienced shivering [8]. A study by Einstein et al. suggested that headaches occur because the mRNA vaccine produced spike proteins that can cross the blood–brain barrier, causing intracranial inflammation [9].

In this study, it was found that there was a significant relationship between the appearance of AEFI symptoms and a previous history of allergies. Desai et al. found that the incidence of anaphylaxis after mRNA vaccine administration was twice as high in recipients with a history of allergy/previous anaphylaxis compared to those without a history (*p* < 0.0001) [10]. Excipients (inactive ingredients used in mRNA vaccines to stimulate a stronger immune response, prevent bacterial contamination, and stabilize vaccine potency during transport and storage) were a major contributor to IgE-mediated and immediate reactions associated with post-vaccination reactions [11].

The presence of AEFI in the first and/or second dose of vaccination influenced the current AEFI occurrence. Our study found significant results at a duration of 24 h–7 days (*p*: 0.009) and 7–28 days (*p* = 0.031). The study conducted by Hause et al. showed that in the Pfizer mRNA vaccine, the subjects who had AEFI in the first dose would have the same AEFI following the second and third doses of the vaccine, varying from local to systemic reactions [6]. Baden et al. conducted a study that compared the Moderna mRNA vaccine and placebo groups, where the mRNA vaccine and placebo were given two doses. The study showed that AEFI at the injection site for the mRNA vaccine group was 84.2% at the first dose, then increased to 88.6% after the second dose. Meanwhile, AEFI in the placebo group decreased from 19.8% in the first dose to 18.8% after the second dose. Systemic AEFIs also showed an increase in the mRNA group, from 54.9% after the first dose to 79.4% after the second dose. Meanwhile, systemic AEFIs in the placebo group decreased from 42.2% after the first dose to 36.5% after the second dose [12]. Meo et al. in their research showed that the degree of AEFI of the Moderna mRNA vaccine was found to be more severe after the second dose than the first dose. At the first dose, the degree of AEFI ranged from mild to moderate, while at the second dose, the degree of AEFI was found to be moderate to severe [13]. Alhaumaid et al. also reported that the risk of AEFI from mRNA vaccine increased in women with a history of previous allergies, due to the formation of polyethylene glycol (PEG) lipid conjugate derivatives. Sensitization to PEG is common in women, because women are relatively more exposed to products containing PEG, such as cosmetics or the use of injectable contraceptive drugs [14].

Our study found that at a duration of <24 h after vaccination, exercise influenced the occurrence of AEFI symptoms. It was found that the group that did not exercise after vaccination was at greater risk of developing AEFI (*p* < 0.003). This finding was in line with a study conducted by Göbel et al., which concluded that 50.7% of 2349 respondents who did regular exercise did not experience post-vaccination headache symptoms [15].

In our study, it was found that for a duration of <24 h after booster vaccination, there was a significant correlation between AEFI occurrence and the age of the recipient. Age below 25 years was found to have a significant effect on AEFI (*p* = 0.005). This finding was in line with a study conducted by the CDC for the Moderna vaccine, which stated that AEFI, especially local reactions, was higher in the young age group (aged 18–64 years) compared to the elderly age group (≥65 years), which was found to be 90.5% vs. 83.9% [16]. The study conducted by Hause et al. also reported the same finding, where respondents > 65 years of age reported experiencing less frequent local or systemic AEFIs than respondents < 65 years old, and local/systemic reactions were reported more frequently after 24 h post-vaccine and decreased up to 7 days post-vaccination [6]. Menni et al., also showed that AEFIs such as headache and fatigue were more common in those <55 years of age compared to >55 years of age [17].

There was a significant relationship between BMI and AEFI symptoms < 24 h following booster vaccination (*p* = 0.03). BMI < 25 kg/m^2^ (underweight/normoweight) was more at risk of experiencing post-vaccination AEFIs than BMI > 25 kg/m^2^ (overweight). This finding is in accordance with a study by Iguacel et al., whichshowed AEFI was higher in the underweight/normal group than in the overweight/obese group [18]. In their study, AEFIs after the second COVID-19 vaccination were found in the underweight group compared to the obese group, such as fever ≥38 °C (25% vs. 10.9%), fever < 38 °C (8.3% vs. 1.8%), myalgia (58.5% vs. 20.0%), swelling and redness in the arm (41.7% vs. 32.7%), headache (58.3% vs. 25.5%), decreased appetite (16.7% vs. 5.5%), sweating (25.0% vs. 3.6%), and shivering (33.3 vs. 10.9%) [18].

This study found that 48 h–7 days after booster vaccination, there was an influence of a previous history of COVID-19 on the appearance of AEFI. This finding is in accordance with the results of a study conducted by Kadali et al., where Moderna vaccine recipients who had a previous history of COVID-19 experienced AEFIs such as shivering (*p* = 0.027), skin redness (*p* = 0.045), tremor (*p* = 0.05), muscle spasms (*p* = 0.039), vomiting (*p* = 0.031), diarrhea (*p* = 0.015), and cough (*p =* 0.011). This group had a higher incidence of AEFIs than the Moderna vaccine group recipients who did not have a previous history of COVID-19 [19].

In this study, it was also found that there were skin rashes that occurred between 48 h and 7 days after vaccination (0.6%) and between 7 and 28 days (1.6%). A similar finding was stated in a study by Johnston et al., which was a slow-type local reaction that appeared within 7 days after receiving the Moderna vaccine [20]. Itching, pain, and/or a red rash were found around the injection site, which Johnston et al. referred to as the “COVID vaccine arm”. According to Johnston, this reaction was a delayed-type hypersensitivity, which was related to the response of T cells to vaccine excipients, nonlipid particles, or mRNA components. Meanwhile, the relationship between polyethylene glycol, which was commonly found in Moderna vaccines, was still not known to be involved in this delayed-type hypersensitivity reaction. According to Johnston, this slow type of local reaction was not a contraindication for the next vaccination [20].

## 5. Conclusions

This study found that there were AEFIs at onset < 24 h to 28 days after the Moderna booster vaccine administrations to health workers of Universitas Indonesia Hospital. The most common AEFI found between <24 and 28 h after vaccination were pain at the injection site, fever, shoulder pain, and headache. The majority of the AEFIs severity level at <24 h to 28 days after booster vaccination was grade 1 (reduced or uninterrupted daily activities). There is a significant correlation between AEFI and several factors, such as a history of drug allergy, exercise, age, BMI < 25, a history of complaints after the first and second vaccinations, and a history of the previous COVID-19. In this study, there was no anaphylactic reaction after administration of the mRNA vaccine (Moderna).

## Figures and Tables

**Table 1 vaccines-10-00877-t001:** Subject characteristics.

Characteristic	N (%)	Median (Range)
Sex		
Male	69 (22.2%)	
Female	242 (77.8%)	
Age (years)		
<25	98 (31.5%)	25 (19–48)
25–39	202 (65%)	
≥40	11 (3.5%)	
BMI (kg/m^2^)		
Underweight (<18.5)	40 (12.9%)	22.86 (15.40–63.29)
Normal weight (18.5–24.9)	173 (55.6%)	
Overweight (≥25)	71 (22.8%)	
Obese (≥30)	27 (8.7%)	
Comorbidity	33 (10,6%)	
History of drug allergy	28 (9%)	
History of food allergy	59 (19%)	
Other history of allergy	60 (19.3%)	
Complains after the 1st or 2nd vaccination	56 (18%)	
History of COVID-19	68 (21.9%)	
Exercise after 3rd vaccination	107 (34.4%)	
AEFI onset < 24 h		
Normal	109 (35%)	
Grade I	161 (51.8%)	
Grade II	41 (13.2%)	
AEFI onset 24–48 h		
Normal	89 (28.6%)	
Grade I	172 (55.3%)	
Grade II	50 (16.1%)	
AEFI onset 48 h–7 days		
Normal	253 (81.4%)	
Grade I	55 (17.7%)	
Grade II	3 (1%)	
AEFI onset 7–28 days		
Normal	301 (96.8%)	
Grade I	8 (2.6%)	
Grade II	2 (0.6%)	

**Table 2 vaccines-10-00877-t002:** Complaints after booster dose.

Complaint	<24 h (%)	24–48 h (%)	48 h–7 Days (%)	7–28 Days (%)
Pain at injection site	265 (85.2%)	240 (77.2%)	96 (31%)	14 (4.6%)
Fever	118 (38%)	158 (50.8%)	14 (4.5%)	6 (1.9%)
Shoulder pain	112 (36.1%)	160 (51.4%)	32 (10.3%)	9 (2.9%)
Headache	13 (4.2%)	104 (33.3%)	16 (5.1%)	-
Sore throat	12 (3.9%)	8 (2.6%)	-	-
Diarrhea	8 (2.6%)	7 (2.3%)	-	-
Skin rash	6 (1.9%)	8 (2.5%)	2 (0.6%)	5 (1.6%)
Feverish	3 (0.9%)	4 (1.3%)	8 (2.6%)	-
Malaise	3 (0.9%)	-	-	-
Shorter breath	3 (0.9%)	4 (1.3%)	-	2 (0.6%)
Sleepy	2 (0.6%)	-		-
Nausea	1 (0.3%)	6 (1.9%)	-	
Myalgia	1 (0.3%)	-	1 (0.3%)	3 (1%)
Tiredness	1 (0.3%)	-	-	-
Itch and bruise at injection site	-	4 (1.3%)	-	-
Cough	-	3 (1%)	5 (1.6%)	5 (1.6%)
Palpitation	-	3 (1%)	-	-
Weakness	-	3 (1%)	-	-
Shivering	-	2 (0.6%)	-	-
Leg pain	-	-	-	2 (0.6%)

**Table 3 vaccines-10-00877-t003:** Factors influencing AEFI in <24 h; 24–48 h; 48 h–7 days; and 7–28 days.

Possible Risk Factors	Severity < 24 h	*p*-Value	RR (95%CI)	Severity 24–48 h	*p*-Value	RR (95%CI)	Severity 48 h–7 days	*p*-Value	RR (95%CI)	Severity 7–28 days	*p*-Value	RR (95%CI)
	Have AEFI/No AEFI n (%)			Have AEFI/No AEFI n (%)			Have AEFI/No AEFI n (%)			Have AEFI/No AEFI n (%)		
**Comorbidities**												
Present	18 (8.9)/15 (13.8)	0.246	0.648 (0.340–1.233)	26 (11.7)/7 (7.9)	0.416	1.498 (0.671–3.306)	6 (10.3)/27 (10.7)	0.980	0.969 (0.420–2.239)	2 (20.0)/31 (10.3)	0.287	1.942 (0.538–7.010)
Not Present	184 (91.1)/94 (86.2)		1.056 (0.969–1.152)	196 (88.3)/82 (92.1)		0.958 (0.887–1.035)	52 (89.7)/226 (89.3)		1.004 (0.911–1.106)	8 (80.0)/270 (89.7)		0.892 (0.653–1.219)
**Drug allergy**												
Present	24 (11.9)/4 (3.7)	0.021 *	3.238 (1.153–9.092)	20 (9.0)/8 (9.0)	1.000	1.002 (0.458–2.191)	7 (12.1)/21 (8.3)	0.443	1.454 (0.649–3.257)	1 (10.0)/27 (9.0)	1.000	1.115 (0.168–7.408)
Not Present	178 (88.1)/105 (96.3)		0.915 (0.859–0.974)	202 (91.0)/81 (91.0)		1.000 (0.925–1.080)	51 (87.9)/232 (91.7)		0.959 (0.866–1.062)	9 (90.0)/274 (91.0)		0.989 (0.802–1.219)
**Food allergy**												
Present	40 (19.8)/19 (17.4)	0.652	1.136 (0.693–1.862)	41 (18.5)/18 (20.2)	0.750	0.913 (0.556–1.500)	7 (12.1)/52 (20.6)	0.192	0.587 (0.281–1.255)	2(20.0)/57 (18.9)	1.000	1.056 (0.299–3.729)
Not Present	162 (80.2)/90 (82.6)		0.971 (0.870–1.084)	181 (81.5)/71 (79.8)		1.022 (0.905–1.155)	51 (87.9)/201 (79.4)		1.107 (0.987–1.241)	8 (80.0)/244 (81.1)		0.987 (0.720–1.352)
**Other allergies**												
Present	44 (21.8)/16 (14.7)	0.136	1.484 (0.880–2.503)	44 (19.8)/16 (18.0)	0.753	1.102 (0.658–1.848)	13 (22.4)/47 (18.6)	0.580	1.207 (0.700–2.078)	0 (0.0)/60 (19.9)	0.218	–
Not Present	158 (78.2)/93 (85.3)		0.917 (0.824–1.020)	178 (80.2)/73 (82.0)		0.978 (0.869–1.099)	45 (77.6)/206 (81.4)		0.953 (0.820–1.107)	10 (100.0)/241 (80.1)		1.249 (1.181–1.321)
**Complaint following previous COVID-19 vaccination**												
Present	42 (20.8)/14 (12.8)	0.054	1.619 (0.926–2.829)	48 (21.6)/8 (9.0)	0.009*	2.405 (1.186–4.878)	16 (27.6)/40 (15.8)	0.031 *	1.745 (1.053–2.890)	6 (60.0)/50 (16.6)	0.003 *	3.612 (2.051–6.360)
Not Present	160 (79.2)/95 (87.2)		0.909 (0.822–1.005	174 (78.4)/81 (91.0)		0.861 (0.783–0.947)	42 (72.4)/213 (84.2)		0.860 (0.727–1.017)	4 (40.0)/251 (83.4)		0.480 (0.224–1.026)
**History of COVID-19**												
Present	46 (22.8)/22 (20.2)	0.667	1.128 (0.718–1.772)	53 (23.9)/15 (16.9)	0.224	1.417 (0.844–2.377)	20 (34.5)/48 (19.0)	0.014 *	1.818 (1.174–2.813)	3 (30.03)/65 (21.6)	0.460	1.389 (0.526–3.668)
Not Present	156 (77.2)/87 (79.8)		0.958 (0.858–1.091)	169 (76.1)/74 (83.1)		0.916 (0.813–1.031)	38 (65.5)/205 (81.0)		0.809 (0.665–0.984)	7 (70.0)/236 (78.4)		0.893 (0.592–1.345)
**Exercise after 3^rd^ dose vaccination**												
Present	57 (28.2)/50 (45.9)	0.003 *	0.615 (0.456–8.30)	158 (71.2)/46 (51.7)	0.001 *	0.597 (0.443–0.804)	40 (69.0)/164 (64.8)	0.646	0.882 (0.581–1.341)	8 (80.0)/196 (65.1)	0.503	0.573 (0.164–1.999)
Not Present	145 (71.8)/58 (54.1)		1.326 (1.093–1.609)	64 (28.8)/43 (48.3)		1.377 (1.108–1.712)	18 (31.0)/89 (32.5)		1.064 (0.875–1.293)	2 (20.0)/105 (34.9)		1.229 (0.891–1.693)
**Age**												
<25 years	75 (37.1)/23 (21.1)	0.005 *	1.760 (1.174–2.638)	75 (33.8)/23 (25.8)	0.180	1.307 (0.879–1.945)	17 (29.3)/81 (32.0)	0.755	0.915 (0.591–1.419)	5 (50.0)/93 (30.9)	0.297	1.618 (0.851–3.076)
>25 years	127 (62.9)/86 (78.9)		0.979 (0.690–0.920)	147 (66.2)/66 (74.2)		0.893 (0.765–1.042)	41 (70.7)/172 (68.0)		1.040 (0.863–1.252)	5 (50.0)/208 (69.1)		0.724 (0.388–1.351)
**BMI**												
<25	147 (72.8)/66 (60.6)	0.030 *	1.202 (1.010–1.429)	155 (69.8)/58 (65.2)	0.422	1.071 (0.900–1.276)	39 (67.1)/174 (68.8)	0.876	0.978 (0.802–1.192)	5 (50.0)/208 (69.1)	0.297	0.724 (0.388–1.351)
≥25	55 (27.2)/43 (39.4)		0.690 (0.499–0.954)	67 (30.2)/31 (34.8)		0.866 (0.612–1.227)	19 (32.8)/79 (31.2)		1.049 (0.695–1.583)	5 (50.0)/93 (30.9)		1.618 (0.851–3.076)
**Gender**												
Male	45 (22.3)/24 (22.0)	1.000	1.012 (0.653–1.567)	49 (22.1)/20 (22.5)	1.000	0.982 (0.621–1.553)	12 (20.7)/57 (22.5)	0.862	0.918 (0.528–1.597)	3 (30.0)/66 (21.9)	0.467	1.368 (0.518–3.611)
Female	157 (77.7)/85 (78.0)		0.997 (0.880–1.128)	173 (77.9)/69 (77.5)		1.005 (0.881–1.147)	46 (79.3)/196 (77.5)		1.024 (0.884–1.186)	7 (70.0)/235 (78.1)		0.897 (0.595–1.351)

* *p*-value < 0.05.

## Data Availability

The data presented in this study are available on request from the corresponding author. The data are not publicly available due to patients’ privacy concerns.

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
