# Peer review of "Surveillance of Adverse Events Following Immunization (AEFI) after Third Dose Booster Vaccination with mRNA-Based Vaccine in Universitas Indonesia Hospital Health Personnel"

_vaccines, 2022, doi:10.3390/vaccines10060877_

Round 1

Reviewer 1 Report

Well conducted study.

I have noticed that those who had high antibody titers to the vaccine develop autoimmune features such as arthritis, mylagia and enhanced degree of dysglycemia. These aspects I wish to discuss i an accompanying editorial or commentary.

Author Response

Dear editor,

thank you for consideration of our manuscript entitled " Surveillance of Adverse Events Following Immunization (AEFI) after Third Dose Booster Vaccination with MRNA-Based Vaccine in Universitas Indonesia Hospital Health Personnel". We have reviewed all the comments, provided revisions, and wished to re-submit our manuscript. 

We hope that the revised version is suitable for publication.

Thank you.

Reviewer 2 Report

This is an interesting study that contributes to the ongoing literature about adverse events after vaccination against COVID-19. The manuscript was easy to follow, although there were several minor mistakes/typos, that I note below for the authors.

Some suggestions that could benefit the manuscript:

Line 37: 13th 2022, 2066 Indonesian healthcare workers died from COVID19.

Here, it would be useful to include the estimated number of all healthcare workers (HCWs) in Indonesia, just to get an idea. Also, if the average age of HCWs in Indonesia is known, then that age group could be compared with the mortality of the general population (for the same age group), just to get an idea how badly the Indonesian HCWs were hit by the pandemic.

Table 3: please correct AFEI. Also, it is not clear what are the two columns for each time slot. Is it male/female?

Line 216-219: It would be more informative to not only mention the percentage, but the relative protection. What was the background frequency for those not doing exercise?

The Discussion was thorough and well documented. It would be a very nice addition if the authors managed somehow to summarize all these findings from the various studies and their study in a table or figure that shows the different percentages of AEFIs in the various studies. Maybe extra columns could be used to make comments about certain studies?

Concerning the Discussion, it would be nice to add a few sentences about the rapid evolution of SARS-CoV-2 and coronaviruses in general, especially in the spike protein, by point mutations and recombination (see DOI: 10.1093/molbev/msab292, DOI: 10.3390/v14040707, DOI: 10.3390/v14010078). Due to this rapid evolution, new variants may emerge that significantly escape the immune response (see DOI: 10.1016/j.cell.2021.03.013, DOI: 10.1016/j.cell.2021.02.037, DOI: 10.1016/j.cell.2021.03.055, DOI: 10.1056/NEJMoa2102214). Furthermore, the highly mutated Omicron variant can escape 26 out of 29 monoclonal antibodies that target the highly mutated Spike receptor binding motif (RBM) (DOI: 10.1038/s41586-021-04386-2). Therefore, it is highly probable that new variants may emerge during the Autumn/Winter of 2022-2023. These new variants will probably be the reason for a fourth dose. Most probably it will be one of the mRNA vaccines that target the rapidly evolving Spike protein (see efficacies in DOI: 10.1093/ofid/ofac138). Therefore, this study is very timely and relevant and may guide the fourth dose, in the future, whether it is the same mRNA or an updated mRNA vaccine.

Line 3: mRNA

Line 14: have received the third dose.

Line 31: COVID19 vaccines have been proven to not only reduce…

Line 30-34: this sentence is too long, better break it in two.

Line 40: Here, it would be useful to be more descriptive, as to which specific vaccines were given as a third dose and maybe at what frequencies.

Table 2: the last line is a repetition of the first/titles

Line 112: That's table 2, not 3, right?

Line 127: and 6 subjects …

Line 131: Respondents who had a history of drug allergy had an increased risk of experiencing… Same for the following sentences

Line 137 and 144: Table 3 also shows that …

Line 171-174: this sentence is too long, please break it in two.

Line 175: the authors mean the Hause study?

Line 179: because mRNA vaccines produce spike proteins…

Also, in that sentence, please include the citation number.

Line 183: Do the authors mean more frequent instead of significant, or more severe?

Line 193-195: Please rephrase, AEFI mentioned too frequently.

Line 200: were given two doses

Line 215: Here, there is a link between exercise and AEFI, but no proven causality. Please correct this throughout the manuscript.

Line 226: The study conducted by Hause…

Line 239: is in accordance

Line 241: please include citation in this sentence.

Line 241: “In this study” of Iguacel?

Line 247-252: this sentence is too long.

Line 258: Please include here the citation of Johnston et al.

Author Response

Dear editor,

thank you for consideration of our manuscript entitled " Surveillance of Adverse Events Following Immunization (AEFI) after Third Dose Booster Vaccination with MRNA-Based Vaccine in Universitas Indonesia Hospital Health Personnel". We have reviewed all the comments, provided revisions, and wished to re-submit our manuscript. 

Reviewer 2

Comments

Line 37: 13th 2022, 2066 Indonesian healthcare workers died from COVID19.

Here, it would be useful to include the estimated number of all healthcare workers (HCWs) in Indonesia, just to get an idea. Also, if the average age of HCWs in Indonesia is known, then that age group could be compared with the mortality of the general population (for the same age group), just to get an idea how badly the Indonesian HCWs were hit by the pandemic.

We already add data about the number of HCWs in Indonesia, as seen in line 34-36, but we can’t find the average age of HCWs.

Table 3: please correct AFEI. Also, it is not clear what are the two columns for each time slot. Is it male/female?

We already changed the AFEI word, and we already change which one that have AEFI and not.

Line 216-219: It would be more informative to not only mention the percentage, but the relative protection. What was the background frequency for those not doing exercise?

In Gobel et al journal, they just mention the percentage and the number of samples. They don’t mention about the relative risk or relative protection. They also don’t explain about the possible cause. We add the number of participants on the manuscript.

The Discussion was thorough and well documented. It would be a very nice addition if the authors managed somehow to summarize all these findings from the various studies and their study in a table or figure that shows the different percentages of AEFIs in the various studies. Maybe extra columns could be used to make comments about certain studies?

Thank you for the suggestions but we don’t think it is necessary since our study is not systematic review nor meta-analysis study. Our references on discussion part also not that much.

Concerning the Discussion, it would be nice to add a few sentences about the rapid evolution of SARS-CoV-2 and coronaviruses in general, especially in the spike protein, by point mutations and recombination (see DOI: 10.1093/molbev/msab292, DOI: 10.3390/v14040707, DOI: 10.3390/v14010078). Due to this rapid evolution, new variants may emerge that significantly escape the immune response (see DOI: 10.1016/j.cell.2021.03.013, DOI: 10.1016/j.cell.2021.02.037, DOI: 10.1016/j.cell.2021.03.055, DOI: 10.1056/NEJMoa2102214). Furthermore, the highly mutated Omicron variant can escape 26 out of 29 monoclonal antibodies that target the highly mutated Spike receptor binding motif (RBM) (DOI: 10.1038/s41586-021-04386-2). Therefore, it is highly probable that new variants may emerge during the Autumn/Winter of 2022-2023. These new variants will probably be the reason for a fourth dose. Most probably it will be one of the mRNA vaccines that target the rapidly evolving Spike protein (see efficacies in DOI: 10.1093/ofid/ofac138). Therefore, this study is very timely and relevant and may guide the fourth dose, in the future, whether it is the same mRNA or an updated mRNA vaccine.

Thank you for the suggestions but we don’t think it is necessary to explain about the evaluation SARS-CoV-2 in that deep knowledge on our discussion. We will openly accept suggestions if it related with the adverse effects.

Line 3: mRNA

We already revised it.

Line 14: have received the third dose.

We already revised it, as seen in line 12 and 13.

Line 31: COVID19 vaccines have been proven to not only reduce…

We already revised it, as seen in line 30 and 31.

Line 30-34: this sentence is too long, better break it in two.

We already revised it.

Line 40: Here, it would be useful to be more descriptive, as to which specific vaccines were given as a third dose and maybe at what frequencies.

The government data just mention about the administration of booster dose. It doesn’t explain which type of vaccine that used on the first, second and third dose.

Table 2: the last line is a repetition of the first/titles

We already revised it.

Line 112: That's table 2, not 3, right?

We already revised it.

Line 127: and 6 subjects …

We already revised it and that sentence have been deleted.

Line 131: Respondents who had a history of drug allergy had an increased risk of experiencing… Same for the following sentences

We already revised it and that sentence have been deleted.

Line 137 and 144: Table 3 also shows that …

We already revised it.

Line 171-174: this sentence is too long, please break it in two.

We already short the sentences, as seen in line 368.

Line 175: the authors mean the Hause study?

We already short the sentences, as seen in line 368.

Line 179: because mRNA vaccines produce spike proteins…

We already revised it, as seen in line 373

Also, in that sentence, please include the citation number.

We already revised it.

Line 183: Do the authors mean more frequent instead of significant, or more severe?

The authors mean more frequent, we already revised it in line 377.

Line 193-195: Please rephrase, AEFI mentioned too frequently.

We already revised it, as seen in line 387.

Line 200: were given two doses

We already revised it, as seen in line 393.

Line 215: Here, there is a link between exercise and AEFI, but no proven causality. Please correct this throughout the manuscript.

We already revised it.

Line 226: The study conducted by Hause…

We already revised it, as seen in line 521

Line 239: is in accordance

We already revised it, as seen in line 534

Line 241: please include citation in this sentence.

We already revised it.

Line 247-252: this sentence is too long.

We already revised it.

Line 258: Please include here the citation of Johnston et al.

We already revised it.

We hope that the revised version is suitable for publication.

Thank you.

Reviewer 3 Report

Health care worker refusal to be vaccinated is an important topic worldwide.

Sample: What is the total number of eligibles? How did you draw up your sampling frame? How were the 311 selected? How many refusals? How many incomplete data (and is so what did you do?)  

Tables: Table 3 is much too detailed, contains many small numbers and must be placed as a Supplemental Table

Conclusions: What is the generalisability of your study? To whom can it be generalised with low risk of bais?

The paper is much too long for the content. Please shorten by 50%.

Typos:

lines 70, 75 data is (Latin; datum = singular, data = plural)

75 311 samples [please change to persons]

88,89 dominated [lease find a neutral term instead] 

Author Response

Dear editor,

thank you for consideration of our manuscript entitled " Surveillance of Adverse Events Following Immunization (AEFI) after Third Dose Booster Vaccination with MRNA-Based Vaccine in Universitas Indonesia Hospital Health Personnel". We have reviewed all the comments, provided revisions, and wished to re-submit our manuscript. 

Reviewer 3

Comments

Health care worker refusal to be vaccinated is an important topic worldwide.

Comment only, no revision needed here.

Sample: What is the total number of eligibles? How did you draw up your sampling frame? How were the 311 selected? How many refusals? How many incomplete data (and is so what did you do?)

We use total sampling method through online questionnaire. The online questionnaire consists of mandatory questions and optional questions. The questionnaire can’t be submitted before the mandatory questions is filled out.

The amount of people who fill out the questionnaire is 311. If there are some missing data or questions that we want to ask, we will reach them through their phone number that they put on the questionnaire.

Tables: Table 3 is much too detailed, contains many small numbers and must be placed as a Supplemental Table

We already re-format the table 3. We hope it can make people understand it easier

Conclusions: What is the generalisability of your study? To whom can it be generalised with low risk of bais?

Our study can be generalised for health workers, but it can be taken for considerations for national booster administration.

The paper is much too long for the content. Please shorten by 50%.

We already decreased the content. The journal still asked for minimum 4000 words for article manuscript.

Typos:

lines 70, 75 data is (Latin; datum = singular, data = plural)

75 311 samples [please change to persons]

88,89 dominated [lease find a neutral term instead]

In line 78, we use data because it means personal data(s) such as their name, age, weight, height, etc.
In line 84, we use data because we put every subject’s data to excel.

We already revised it, as seen in line 84.

We already revised it, as seen in line 97 and 98.

We hope that the revised version is suitable for publication.

Thank you.

Round 2

Reviewer 3 Report

Thanks to the authors for their changes